# MGST1 Protects Pancreatic Ductal Cells from Inflammatory Damage in Acute Pancreatitis by Inhibiting Ferroptosis: Bioinformatics Analysis with Experimental Validation

**DOI:** 10.3390/ijms26051899

**Published:** 2025-02-22

**Authors:** Ruoyi Zhang, Xin Ling, Xianwen Guo, Zhen Ding

**Affiliations:** Department of Gastroenterology, The First Affiliated Hospital, Sun Yat-Sen University, Guangzhou 510080, China; zhangruoyi_dsa@163.com (R.Z.);

**Keywords:** acute pancreatitis, ferroptosis, immune cells, pancreatic ductal cells

## Abstract

Numerous animal experiments have implicated ferroptosis in the pathogenesis of acute pancreatitis (AP). Nonetheless, due to sampling constraints, the precise role of ferroptosis in the human body during AP remains elusive. Method: Peripheral blood sequencing data of patients with acute pancreatitis (GSE194331) were obtained from the Gene Expression Omnibus (GEO) database. We analyzed differentially expressed genes whose expression increased or decreased with increasing disease severity and intersected them with the ferroptosis gene set to identify ferroptosis-related driver genes for the disease. The hub genes were selected using machine learning algorithms, and a nomogram diagnosis model was constructed. Clinical samples, animal models, and an in vitro experiment were also used for validation. The investigation unveiled 22 ferroptosis-related driver genes, and we identified three hub genes, *AQP3*, *TRIB2*, and *MGST1*, by employing two machine learning algorithms. *AQP3* and *TRIB2* exhibit robust correlations with various immune cells. The disease diagnosis model constructed utilizing these three genes demonstrated high sensitivity and specificity (AUC = 0.889). In the in vitro experiments, we discovered for the first time that ferroptosis occurs in pancreatic duct cells during acute pancreatitis, and that MGST1 is significantly upregulated in duct cells, where it plays a crucial role in negatively regulating ferroptosis via the ACSL4/GPX4 axis. In addition, overexpression of MGST1 protects ductal cells from inflammatory damage. In our investigation, we explored the mechanisms of ferroptosis in immune cells and pancreatic duct cells in patients with AP. These results highlight a potential pathway for the early diagnosis and treatment of acute pancreatitis.

## 1. Introduction

Acute pancreatitis (AP) is marked by sudden inflammation of the pancreas and histological damage to acinar cells. It ranks among the common emergencies in the digestive system, frequently progressing to severe pancreatitis through local development that affects systemic organs and systems. AP globally affects 130,000 to 450,000 individuals annually. Of these, 80 to 85% experience mild and moderate AP with a self-limiting course, while about 20% progress to severe forms of the disease, with a mortality rate ranging from 13 to 35% [1].

The complete understanding of the pathogenesis of AP remains elusive. In addition to aberrant trypsinogen activation, the pathogenesis of acute pancreatitis is intricately linked with mitochondrial dysfunction, endoplasmic reticulum stress, and impaired autophagy [2]. Presently, effective treatment modalities for acute pancreatitis are lacking, presenting a formidable challenge in mitigating pancreatic function loss [3]. Consequently, elucidating the pathophysiological intricacies of acute pancreatitis is imperative for identifying promising therapeutic avenues.

Ferroptosis is a finely regulated form of programmed cell death distinguished by its iron-dependent buildup of lipid peroxidation to lethal thresholds [4]. The inhibition of cystine transport protein leads to intracellular glutathione depletion, consequently incapacitating GPX4 (glutathione peroxidase 4) and fostering lipid peroxidation accumulation, thereby prompting cell demise. Sensitivity to ferroptosis is intricately intertwined with a multitude of biological processes, spanning iron and polyunsaturated fatty acid metabolism, as well as the synthesis pathways of glutathione and phospholipids [5]. Ferroptosis is linked to various immune diseases, including nonalcoholic steatohepatitis [6], diabetes [7], neuroinflammation [8], and rheumatoid arthritis [9]. Emerging evidence suggests that ferroptosis may trigger harmful inflammatory responses, ultimately leading to cellular membrane disruption and the release of damage-associated molecular patterns (DAMPs) [10].

Recent studies have implicated ferroptosis in the pathogenesis of acute pancreatitis. For instance, one study demonstrated that trypsin activates the proteasome 26S subunit, thereby promoting lipid peroxidation and cell death in acinar cells through GPX4 degradation [11]. Recent studies have shown that the presence of circulating SQSTM1 (sequestosome 1) protein leads to the upregulation of ACSL4 (acyl-CoA synthetase long-chain family member 4) expression in acinar cells, a process that is dependent on the activation of AGER (advanced glycation end-products receptor) [12]. This induction leads to the synthesis of polyunsaturated fatty acids, facilitating autophagosome formation and the subsequent initiation of ferroptosis. However, investigations into the mechanism of ferroptosis in acute pancreatitis are presently constrained to animal experiments, primarily due to limitations in sample acquisition. Consequently, the involvement of ferroptosis in the pathophysiology of acute pancreatitis in humans remains poorly understood. Notably, a recent clinical study unveiled serum iron and lactate levels as independent prognostic factors for mortality in acute pancreatitis patients [13]. These associations imply that ferroptosis could hold noteworthy clinical relevance within this patient population.

This study aimed to elucidate the mechanism of ferroptosis in disease by conducting a bioinformatics analysis of peripheral blood sequencing data obtained from patients diagnosed with acute pancreatitis. Furthermore, it sought to identify hub genes that regulate ferroptosis as potential biomarkers for disease diagnosis. The findings were validated using clinical samples obtained from patients with acute pancreatitis of varying severity, as well as through animal models. We then conducted in vitro experiments to verify that one of the hub genes exhibited significant expression differences in human pancreatic ductal cells during disease. Additionally, we investigated its causal relationship with ferroptosis by knocking down and overexpressing this gene. The flow chart can be seen in Figure 1.

## 2. Results

### 2.1. Identification of DEGs and Ferroptosis-Related Driving Genes in Acute Pancreatitis

Samples from the GSE194331 dataset were stratified into four groups: healthy control, mild AP, moderately severe AP, and severe AP. In the principal component analysis (PCA) plot (Figure 2A), the sample distribution of the four groups shows that the healthy control group samples are predominantly concentrated on the left side of the plot, while the severe AP group samples are primarily located on the right side. The samples from the mild AP and moderately severe AP groups are mostly situated in the central region, and the distribution of these two groups is relatively close, indicating a high similarity in molecular characteristics between mild AP and moderately severe AP. This finding is consistent with clinical observations, where the majority of mild AP and moderately severe AP patients typically exhibit a self-limiting disease course [14], characterized by slow disease progression and recovery through supportive treatment. Based on this molecular similarity and in consideration of clinical features, we optimized the sample grouping to simplify the analysis and enhance the specificity of this study. The mild AP and moderately severe AP groups were combined into the “mild to moderately severe AP” group, and the samples were reclassified into three groups: healthy control, mild to moderately severe AP, and severe AP. Figure 2B presents the PCA plot of the three newly reclassified groups. In this plot, the three groups show significant distribution differences along the PCA coordinates, further validating the rationale for the reclassification and providing clearer molecular characteristic differences for subsequent research.

To identify the differentially expressed genes (DEGs) associated with disease severity, we first conducted gene expression analysis between the healthy control group and the mild to moderately severe AP group. A total of 1929 genes were significantly upregulated in the mild to moderately severe AP group, while 1419 genes were significantly downregulated. Figure 2C displays the heatmap of the top 50 most differentially expressed genes, including the 25 most upregulated and 25 most downregulated genes, along with the volcano plot depicting all differentially expressed genes. In the comparison between the mild to moderately severe AP group and the severe AP group, 641 genes were significantly upregulated and 366 genes were significantly downregulated in the severe AP group. Figure 2D shows the heatmap of the top 50 most differentially expressed genes, comprising the 25 most upregulated and 25 most downregulated genes from both groups, accompanied by the volcano plot for all differentially expressed genes.

Among these, 410 genes were upregulated in the mild to moderately severe AP group compared to the healthy control group and further upregulated in the severe AP group compared to the mild to moderately severe AP group. Additionally, 159 genes were downregulated in the mild to moderately severe AP group compared to the healthy control group and further downregulated in the severe AP group compared to the mild to moderately severe AP group. As a result, we identified 410 DEGs that were progressively upregulated with increasing disease severity and 159 DEGs that were progressively downregulated as the disease severity increased.

Subsequently, we intersected the 410 DEGs that were upregulated with increasing disease severity with the 484 ferroptosis-related genes from the FerrDb v2 database, identifying 20 ferroptosis-related DEGs that were upregulated with disease severity (Figure 2E). Additionally, we intersected the 159 DEGs that were downregulated with increasing disease severity with the ferroptosis gene set, resulting in two ferroptosis-related DEGs that were downregulated with disease severity (Figure 2F). These 22 genes were referred to as the ferroptosis driver genes in AP. The results are presented using Venn diagrams.

### 2.2. Enrichment Analysis of Ferroptosis-Related Driving Genes and Correlation Analysis

Gene ontology (GO) enrichment analysis and KEGG pathway database analysis were employed to categorize the biological functions of the 22 ferroptosis-related driver genes. Kyoto Encyclopedia of Genes and Genomes (KEGG) pathway analysis revealed that these genes were significantly enriched in pathways such as glutathione metabolism, xenobiotic metabolism via cytochrome P450, and carbon metabolism (Figure 3A). In terms of biological processes (BP), the most significantly enriched GO terms included response to oxidative stress, neutrophil degranulation, and neutrophil activation (Figure 3B). For the cellular components (CC) category, these genes were primarily localized in peroxisomes and secretory granules (Figure 3C). Regarding molecular functions (MF), these genes were mainly involved in the regulation of oxidoreductase activity and NADP binding (Figure 3D).

We performed a correlation analysis to investigate the relationship between these 22 ferroptosis-related driver genes. The results indicated that the expression levels of 20 genes were positively correlated with each other. However, *AQP3* (aquaporin 3) and *TRIB2* (tribbles pseudokinase 2) showed negative correlations with the expression levels of the other 20 genes. The results are presented as a correlation heatmap in Figure 3E. Figure 3F presents a heatmap of the expression levels of the 22 ferroptosis driver genes across the healthy control group, the mild to moderately severe AP group, and the severe AP group. Among these genes, 20 exhibited increased expression with increasing disease severity, while *AQP3* and *TRIB2* showed decreased expression as the disease severity progressed.

### 2.3. Identification of Hub Genes Using Machine Learning Methods and Construction of a Diagnostic Model

In this study, we employed two widely recognized machine learning algorithms, least absolute shrinkage and selection operator (LASSO) regression and support vector machine (SVM), to identify ferroptosis-related hub genes implicated in the progression of acute pancreatitis. First, LASSO regression was applied to the 22 ferroptosis-related driver genes, with the optimal regularization parameter (λ) determined through 10-fold cross-validation. As a result, five candidate genes were selected. The LASSO path plot typically shows how binomial deviance changes with different values of the regularization parameter λ (Figure 4A), further validating the stability of the selection process. Additionally, the SVM algorithm was used to identify hub genes associated with acute pancreatitis severity. After optimizing hyperparameters through 10-fold cross-validation, SVM identified six candidate genes (Figure 4B). By integrating the results from both LASSO and SVM analyses, five and six candidate genes were identified, respectively. Intersection analysis revealed three hub genes—*AQP3*, *TRIB2*, and *MGST1* (microsomal glutathione S-transferase 1) (Figure 4C).

Gene expression analysis showed a decreasing trend in the expression levels of *AQP3* and *TRIB2* with increasing disease severity, while the expression level of *MGST1* significantly increased. Pairwise comparisons between groups revealed that the expression differences of these genes across the different disease groups were statistically significant (*p* < 0.01) (Figure 4D–F).

Gene set enrichment analysis (GSEA) was performed to delve deeper into the potential molecular mechanisms underlying these three hub genes. Figure 4G–I present the top 10 enrichment KEGG terms in the high- or low-expression groups of three hub genes. Several immune-related pathways emerged as significant, including toll-like receptor signaling, nucleotide-binding oligomerization domain-like receptor (NOD) signaling, Fc gamma-mediated phagocytosis, immunodeficiency, chemokine signaling, and leukocyte transendothelial migration.

Through receiver operating characteristic (ROC) curve analysis, we found that all three hub genes demonstrated excellent diagnostic performance, with the AUC for each gene exceeding 0.8. Additionally, the combined model of the three hub genes yielded an overall area under curve (AUC) value of 0.889 (Figure 4J), suggesting the potential diagnostic value of these genes for acute pancreatitis. To enhance the clinical applicability of the model, we further constructed a nomogram prediction model based on these three hub genes (Figure 4K). Calibration curve analysis showed a high degree of agreement between the predicted probabilities and the observed probabilities (Figure 4L), indicating strong predictive ability. The slope of the calibration curve was close to 1, further confirming the accuracy and stability of the model (Figure 4M).

### 2.4. Association Between Hub Genes and Immune Cell Infiltration

Using the CIBERSORT algorithm, we conducted a systematic analysis of the immune cell composition differences between AP patients and healthy controls. Figure 5A presents the proportion distribution of 22 immune cell subtypes between the AP patients and the healthy control group. Since the expression of six immune cell types was zero in both the healthy control and disease groups, Figure 5B further analyzes the expression trends of the remaining 16 immune cell types across the healthy control group, mild to moderately severe AP group, and severe AP group.

The analysis revealed that, compared to the healthy control group, the infiltration levels of resting CD4+ T cells, memory B cells, CD8+ T cells, and activated natural killer (NK) cells were significantly reduced in the peripheral blood of AP patients. In contrast, the infiltration levels of M0 macrophages and neutrophils were significantly increased in AP patients. Additionally, as disease severity increased, the infiltration ratio of resting CD4+ T cells in the peripheral blood of AP patients exhibited a gradual decrease. The lollipop chart demonstrates the significant correlations between the three hub genes and various immune cells (Figure 5C–E).

Correlation analysis results indicated that among immune cells involved in the adaptive immune response, *AQP3* and *TRIB2* showed significant positive correlations with resting CD4+ T cells, memory B cells, and CD8+ T cells, with the strongest correlation observed with resting CD4+ T cells (Spearman correlation coefficient > 0.8, *p* < 0.05). In contrast, *MGST1* was significantly negatively correlated with resting CD4+ T cells, memory B cells, and CD8+ T cells, and the correlation of *MGST1* with multiple immune cells was relatively weak (Spearman correlation coefficients ranged from 0.01 to 0.7, *p* < 0.05). The correlations between the three core genes and various immune cells, along with their specific correlation coefficients, are shown in Figure 5F,G.

### 2.5. miRNA, lincRNA, and TF Networks of Hub Genes

Interactions between miRNAs, lincRNAs, transcription factors (TFs), and genes were collected using network analysis. Hub genes were screened to identify their interactions with miRNAs, lincRNAs, and TFs. *AQP3* is regulated by three miRNAs and thirteen TFs, *TRIB2* is regulated by one miRNA, one lincRNA, and eleven TFs, and *MGST1* is regulated by one miRNA and twelve TFs, revealing their interconnected regulatory relationships (Figure 6A). To enhance the precision of our investigation, we used the Drug Gene Interaction Database to identify drugs with strong interaction scores related to *AQP3*, *TRIB2,* and *MGST1*, revealing potential therapeutic agents targeting these genes (Figure 6B).

### 2.6. Potential Roles of the Three Hub Genes in Pancreatic Tissue Inferred Through Single-Cell Analysis

The single-cell dataset GSE188819 encompasses a total of 29,071 cells derived from mouse pancreas samples. Dimensionality reduction using t-distributed stochastic neighbor embedding (t-SNE) was applied to aid in data visualization and exploration. Subsequently, the cells were annotated into 12 distinct cell types. Figure 6C depicts the distribution of cells originating from different cell types and sample origins. Figure 6D–F illustrate the specific enrichment patterns of the three hub genes across different cell types within pancreatic tissue. Specifically, *AQP3* is primarily enriched in NK T cells within pancreatic tissue, while *TRIB2* is predominantly enriched in B cells. Conversely, the expression of *MGST1* was heightened in pancreatic duct cells. These findings suggest that *AQP3* and *TRIB2* may modulate the pancreatic immune microenvironment, while *MGST1* likely plays a pivotal role in the pathological alterations affecting pancreatic ductal cells during disease progression.

### 2.7. Validation of Hub Genes Through Clinical Samples and Animal Models

To ascertain the precise expression levels of three hub genes in AP, peripheral blood mononuclear cells (PBMCs) were procured and subjected to RT-qPCR analysis from 40 clinical AP patients, comprising 24 with mild to moderately severe AP and 16 with severe AP, in addition to PBMCs obtained from 12 healthy volunteers. Clinical data unveiled a gradual decrement in both the absolute count and percentage of lymphocytes with ascending disease severity, with this distinction being most conspicuous among all peripheral blood indicators (Figure 7A). RT-qPCR analysis revealed a decrement in the expression levels of *AQP3* and *TRIB2* with escalating disease severity. However, the expression level of *MGST1* did not increase with the progression of disease severity, suggesting that *MGST1* is not upregulated in the immune cells of human peripheral blood during the disease state (Figure 7B–D).

Subsequent exploration into the expression of hub genes in pancreatic tissue was undertaken using animal models. Histopathological examination disclosed structural destruction of the pancreas and diffuse lobular widening in the AP group (Figure 7E), indicative of the successful establishment of the AP model. Furthermore, RT-qPCR analysis showed a decrease in the expression levels of *AQP3* and *TRIB2*, while *MGST1* expression was significantly elevated in the disease group (Figure 7F–H). Simultaneously, results from Western blot (WB), immunohistochemistry (IHC), and other methods demonstrated that MGST1 protein levels were significantly elevated in the disease state. Based on cell morphology, it was inferred that MGST1 predominantly localized in the exocrine cells (Figure 7I–K). Given the low baseline expression of AQP3 and TRIB2, we were unable to confirm changes in the protein levels of these two genes. To further elucidate the role of MGST1 in acute pancreatitis, our subsequent study will concentrate on the role of MGST1 in the pancreas during the disease state.

### 2.8. MGST1 Is Upregulated During Ferroptosis in Human Pancreatic Duct Cells

To determine the effects of ferroptosis on MGST1 expression, we first treated pancreatic cancer cells with a ferroptosis inducer. In CFPAC-1 cells, protein expression of MGST1 was upregulated in response to RSL3, an effect that was inhibited by liproxstatin-1. Consistent with previous studies, RSL3 treatment resulted in upregulation of the ferroptosis promoter ACSL4 and a downregulation of GPX4 (Figure 8A); these effects were reversed following pretreatment with ferroptosis inhibitors (Figure 8B). However, the expression of MGST1 in mouse acinar cancer cells could not be induced by RSL3. These results suggest that MGST1 may play a role in regulating ferroptosis in pancreatic duct cells rather than acinar cells. Next, we utilized taurocholate sodium (TCS) to replicate the disease environment associated with acute pancreatitis. It is possible that the insensitivity of tumor cells to this drug resulted in the unchanged expression of ferroptosis signature genes in CFPAC-1 cells, with no increase in MGST1 expression.

Then, we used TCS to stimulate normal human ductal cells, and as anticipated, the protein levels of MGST1 were markedly increased in the disease group. Concurrently, ACSL4 levels increased, while NFE2L2 (nuclear factor, erythroid 2 like 2) and GPX4 levels decreased, indicating the occurrence of ferroptosis and GPX4-dependent classical signaling pathway activation in H6C7 cells (Figure 8C). As a positive control, MGST1 levels also significantly increased in H6C7 cells stimulated by the ferroptosis inducer. Concurrently, the GPX4-dependent ferroptosis signaling pathway was activated alongside the AIFM2 (apoptosis-inducing factor mitochondria-associated 2)-dependent non-classical ferroptosis signaling pathway (Figure 8D). These results indicate that MGST1 levels are significantly elevated in pancreatic duct cells during acute pancreatitis, suggesting that MGST1 may regulate ferroptosis in these cells through a GPX4-dependent signaling pathway. Moreover, the protein expression level of MGST1 in the ductal cell supernatant was significantly elevated, suggesting that MGST1 can be secreted into the extracellular space under the influence of TCS (Figure 8E).

In addition, the changes in ferroptosis-related genes induced by TCS in H6C7 cells were reversed following pretreatment with ferroptosis inhibitors (Figure 8F). Meanwhile, cell activity and inflammation levels were restored (Figure 8G–J), suggesting the potential value of ferroptosis inhibitors in treating ductal cell damage associated with diseases.

### 2.9. MGST1 Negatively Regulates Ferroptosis and Protects Human Pancreatic Ductal Cells from Inflammatory Damage

To clarify the specific regulatory effect of MGST1 on ferroptosis in normal human duct cells, we observed that ACSL4 expression was upregulated while GPX4 expression was downregulated in the disease group following MGST1 knockdown using siRNA (Figure 9A,B). Figure 9C–E present the results of protein quantification. Meanwhile, cell viability and GSH/GSSG levels significantly decreased after MGST1 knockdown (Figure 9F,G). These results suggest that TCS-induced ferroptosis in pancreatic ductal cells is aggravated following MGST1 knockdown, indicating that MGST1 may negatively regulate ferroptosis through the ACSL4/GPX4 axis. The levels of various inflammatory factors, including *IL-6*, *IL-1β*, *IL-8*, and *IL-18*, were significantly increased in pancreatic duct cells of the disease group following MGST1 knockdown (Figure 9H–K). The ROS fluorescence intensity increased markedly after MGST1 knockdown during the disease state (Figure 9L,M).

Given that MGST1 is a detoxification enzyme involved in oxidative stress and negatively regulates ferroptosis, we overexpressed MGST1 by transfecting a plasmid into pancreatic duct cells (Figure 10A,B). In the disease group, MGST1 overexpression led to decreased ACSL4 expression and increased GPX4 expression (Figure 10C–E). Furthermore, MGST1 overexpression significantly enhanced cell viability and the GSH/GSSG ratio (Figure 10F,G). As a result, the mRNA expression levels of various inflammatory factors were notably reduced in the disease group following MGST1 overexpression. (Figure 10H–K). These results suggest that MGST1 overexpression can alleviate ferroptosis and inflammatory damage in ductal cells through the ACSL4/GPX4 axis.

## 3. Discussion

Acute pancreatitis is a disease characterized by impaired pancreatic function and immune dysfunction [15]. Key determinants of prognosis in acute pancreatitis include the systemic inflammatory response syndrome (SIRS) and the compensatory anti-inflammatory response syndrome (CARS) [16].

Firstly, we analyzed the differentially expressed genes in acute pancreatitis of varying severity. These genes were then intersected with the ferroptosis gene dataset, resulting in 22 ferroptosis-related driving genes. Subsequently, we employed two machine learning algorithms to identify three hub genes from these 22 genes: *AQP3*, *TRIB2*, and *MGST1*. The disease diagnosis model constructed using these three genes exhibits high sensitivity and specificity. Additionally, *AQP3* and *TRIB2* are strongly associated with various immune cells, particularly T cells, which are involved in the regulation of adaptive immunity. These findings collectively suggest that *AQP3* and *TRIB2* may play a vital role in immune dysregulation and redox status in diseases. Subsequently, we observed that *AQP3* and *TRIB2* were primarily enriched in pancreatic immune cells, whereas MGST1 was primarily enriched in pancreatic duct cells. This suggests that *AQP3* and *TRIB2* may also play a role in regulating the immune microenvironment of pancreatitis. *MGST1* may also be involved in the regulation of pancreatic duct cell damage in the disease state.

Since GSE194331 is the only publicly available human peripheral blood transcriptome dataset for AP, and due to limitations in available tissue samples, there is currently no transcriptome dataset for pancreatic tissue from AP patients. Therefore, we collected PBMCs from patients with varying severities of acute pancreatitis as an external validation cohort. Notably, in line with the results gleaned from our bioinformatics analyses, we noted a progressive reduction in the expression levels of *AQP3* and *TRIB2* as disease severity increased. However, *MGST1* expression was not elevated in the disease group. These results suggest that during acute pancreatitis, the expression of *AQP3* and *TRIB2* in human peripheral blood immune cells gradually decreases with increasing disease severity, whereas *MGST1* does not exhibit differential expression in immune cells. In the animal experiments, we detected a marked elevation in MGST1 protein levels in the pancreatic tissues of mice. Given that we could not confirm the upregulation of MGST1 expression in human peripheral blood immune cells, coupled with the significantly elevated protein expression of MGST1 in diseased pancreatic tissue, we hypothesize that MGST1 is released from pancreatic tissue into the peripheral blood during disease progression.

AQP3 functions as an essential mediator of water and glycerol transport. Notably, research has underscored AQP3′s pivotal role in actin polymerization and subsequent T cell chemotaxis [17]. AQP3 deficiency has been associated with impaired H2O2 permeability and compromised migration of T cells. Furthermore, AQP3 plays a crucial role in facilitating water and glycerol transport, thereby augmenting the phagocytic and migratory activity of macrophages [18]. Inhibition of AQP3 also blocks NLRP3 (NLR family pyrin domain containing 3) inflammasome activation in macrophages [19]. Furthermore, AQP3 participates in modulating dendritic cell population and migration [20]. Overall, AQP3 plays a crucial role in both innate and adaptive immunity.

TRIB2 serves as a novel regulator in various cellular processes. It has been established that TRIB2 exerts regulatory control over cellular proliferation during thymocyte development and in response to genotoxic stress [21]. Notably, the absence of TRIB2 renders cells hypersensitive to genotoxic stress, expediting the onset of T cells. Additionally, TRIB2 is implicated in distinct roles within mature T cell biology and function. Deregulation of TRIB2 can also result in the inactivation of monocytes via the MAPK (mitogen-activated protein kinase) pathway [22]. Furthermore, TRIB2 plays a pivotal role in modulating redox homeostasis [23].

Based on the extensive literature indicating that AQP3 and TRIB2 are predominantly distributed in immune cells and play vital protective roles in various immune cell types, combined with their strong positive correlations with multiple immune cells during AP, we hypothesize that the reduced expression of AQP3 and TRIB2 is a key contributor to immune dysfunction in AP. Furthermore, RT-qPCR results from human peripheral blood immune cells further confirm the decreased expression of AQP3 and TRIB2 in these immune cells, suggesting their potential protective roles in immune cell function.

However, limited studies have explored the effects of MGST1 on immune cells, and we were unable to confirm the upregulation of MGST1 expression in human peripheral blood immune cells. These findings suggest that MGST1 may not exert its effects within immune cells in AP. In contrast, the high expression of MGST1 observed in the pancreatic tissue of AP mice warrants further investigation. Our study subsequently focused on the functional role of MGST1 in diseased pancreatic tissue, aiming to clarify the reasons behind its increased expression during AP and its potential impact on disease progression.

In the cell experiments, a ferroptosis inducer was used to stimulate human pancreatic ductal cancer cells and normal pancreatic ductal cells. This resulted in a significant increase in MGST1 protein levels, accompanied by changes in ferroptosis marker genes, suggesting that MGST1 may play a role in regulating the ferroptosis process in pancreatic ductal cells. Next, we used TCS to simulate the environment of acute pancreatitis. MGST1 protein levels were significantly increased in normal human ductal cells, and activation of the GPX4-dependent classical ferroptosis pathway occurred, mirroring the results observed after treatment with the ferroptosis inducer, and the effect was reversed after pretreatment with ferroptosis inhibitors. Combined with findings from single-cell analysis, we found that changes in MGST1 expression could not be induced in mouse acinar cells. We conclude that MGST1 expression is significantly elevated in human ductal cells rather than in acinar cells during disease progression and MGST1 may regulate ferroptosis through the GPX4-dependent classical signaling pathway. Furthermore, ferroptosis inhibitors may hold potential value in alleviating inflammation in pancreatic duct cells. Under TCS stimulation, MGST1 expression was significantly elevated in the supernatant of ductal cells, further suggesting that MGST1 can be released from ductal cells into the periphery during the onset of acute pancreatitis.

After knocking down MGST1 in normal pancreatic duct cells, we found that the severity of ferroptosis in the disease group was exacerbated, indicating that MGST1 might have a negative regulatory effect on ferroptosis via the ACSL4/GPX4 axis. Subsequently, we overexpressed MGST1 in the duct cells and observed that inflammation levels in the disease group were alleviated. This finding aligns with results from several previous studies indicating that MGST1 is reductive and exerts a protective effect on cells during oxidative stress [24,25,26].

Currently, research on acute pancreatitis primarily focuses on acinar cells, with most studies using mouse-derived acinar cells due to material limitations. However, growing evidence indicates that the pancreatic duct cells are key targets of stressors, possibly contributing significantly to the worsening of acute pancreatitis [27,28]. Prior studies have demonstrated that the primary role of the duct cells is the secretion of fluid and bicarbonate (HCO_3_^−^), which are crucial for digestion. Additionally, proper secretion of duct cells is essential for safeguarding the pancreas against acute pancreatitis [29,30,31]. Nonetheless, few studies have examined the molecular mechanisms underlying pancreatic duct cell injury in acute pancreatitis. Since pancreatic ductal cells have a strong exocrine function, and our cell experiments have confirmed that MGST1 can be released into the extracellular space by ductal cells in the acute pancreatitis environment, we can hypothesize that during acute pancreatitis, MGST1 expression is significantly elevated in ductal cells and released into the peripheral blood.

In this study, normal human pancreatic duct cells were utilized, revealing for the first time that ferroptosis occurs in duct cells during acute pancreatitis. MGST1 is an antioxidant enzyme predominantly located in the mitochondrial outer membrane, endoplasmic reticulum, and peroxisome. It participates in cellular defense against various toxins, oxidative stress, and apoptosis [32]. Notably, MGST1 has been identified as a potential prognostic marker for poor outcomes in patients with pancreatic ductal adenocarcinoma (PDAC) and several other tumors [33,34,35]. It has been demonstrated that MGST1 exerts its inhibitory effect on ferroptosis, in part through its interaction with ALOX5 (arachidonate 5-lipoxygenase), which reduces lipid peroxidation [25]. Our study demonstrated for the first time that MGST1 levels were significantly elevated in the disease state and demonstrated its key negative regulatory role in ferroptosis within pancreatic duct cells. Overexpression of MGST1 was found to alleviate inflammation in duct cells, suggesting a novel target for the treatment of AP.

However, our study has some limitations. Due to sampling constraints, we were unable to validate the hub genes in human pancreatic tissue or assess their potential as prognostic markers for the disease. Further research is necessary to explore the additional mechanisms underlying the functions of these three hub genes in the context of acute pancreatitis.

## 4. Materials and Methods

### 4.1. Data Source and Analysis of Differential Expression Genes (DEGs) 

Peripheral blood sequencing data of patients with acute pancreatitis (GSE194331) were retrieved from the GEO database. GSE194331 comprised 57 patients with mild AP, 20 patients with moderately severe AP, 10 patients with severe AP, and 32 healthy controls. Additionally, GSE188819 was used for single-cell analysis of pancreatic tissue in mice with acute pancreatitis. The FerrDb V2 database includes 369 driver genes, 348 suppressor genes, and 11 marker genes. After removing duplicate annotations, a total of 484 ferroptosis-related genes were identified. Principal component analysis (PCA) was used to reduce the dimensionality of the original variables, transforming them into a new set of components. DEGs were identified using the “limma” package in R (version 4.3.1, R Foundation for Statistical Computing, Vienna, Austria), with statistical significance defined by a *p*-value < 0.05 and |logFC| > 0.5. The selection of the disease dataset is listed in Table 1.

### 4.2. Analysis of Functional Enrichment

This analysis aimed to elucidate the biological relevance and potential functions of the identified genes. To achieve this, the genes underwent gene ontology (GO) analysis, which encompasses biological processes (BP), cellular components (CC), and molecular functions (MF), along with Kyoto Encyclopedia of Genes and Genomes (KEGG) pathway analysis.

### 4.3. Identification of Hub Genes

Two machine learning algorithms were utilized to predict disease status and identify key feature genes. LASSO regression, a regularization-based regression technique, enhances prediction accuracy by selecting important features while reducing overfitting. In contrast, SVM-RFE (support vector machine-recursive feature elimination), a widely used supervised machine learning method, is commonly applied for classification and regression tasks by iteratively removing less relevant features to improve model performance.

### 4.4. ROC Curve Analysis and Diagnostic Model Construction

In the analysis conducted using the “pROC” package in R (version 4.3.1), receiver operating characteristic (ROC) curve analysis was conducted to assess the diagnostic potential of the hub genes. Hub genes with areas under the curve (AUC) greater than 0.7 were deemed potentially valuable for disease diagnosis. The diagnostic model for the disease, based on hub genes, was constructed using a nomogram, which was visualized with the “rms” package in R (version 4.3.1).

### 4.5. Gene Set Enrichment Analysis (GSEA)

The GSEA algorithm compares the distribution of the hub genes within the dataset with the predefined gene sets from MSigDB, assessing whether the hub genes are significantly enriched in specific biological pathways or processes.

### 4.6. Analysis of Immune Cell Infiltration and Its Correlation with Hub Genes

The CIBERSORT method was chosen to compute the 22 immune infiltrating cell scores for each sample based on our expression profiles, utilizing the R-package IOBR. To explore potential associations between infiltrating immune cells and hub genes, Spearman correlation analysis was performed. Additionally, a heatmap illustrating the correlation between infiltrating immune cells and hub genes was generated using the “corrplot” function in the R (version 4.3.1) package.

### 4.7. Construction of TF and miRNA and lncRNA Hub Gene Regulatory Networks

MiRNA and lncRNA targeting hub genes were identified by accessing online datasets such as Starbase and TargetScan for miRNA and SpongeScan for lncRNA and by querying these databases. Additionally, upstream transcription factors (TFs) regulating hub genes were predicted using NetworkAnalyst, a bioinformatics tool for network-based analysis. The predicted regulatory interactions were visualized using Cytoscape software (version 3.9.1).

### 4.8. Single-Cell Analysis

We utilized the GSE188819 dataset, a single-cell RNA sequencing dataset of pancreatic tissue from mice with acute pancreatitis, to analyze changes in the expression of hub genes across different pancreatic cell types. For the analysis, we employed various R (version 3.9.1) packages, including “Seurat” and “Matrix”.

### 4.9. Animals Models

In the experimental protocol, cerulein (HY-A0190, MCE, Shanghai, China) was administered every 1 h at a dose of 50 µg/kg per injection, with a total of seven injections administered throughout the day. Cerulein was diluted with saline for administration. A group of mice (n = 4) received cerulein injections, while a control group (n = 4) received equivalent volumes of normal saline injections at corresponding times. Mice were euthanized by spinal dislocation 24 h after the final injection. We collected pancreatic tissue samples from each mouse for further analysis.

### 4.10. Peripheral Blood Mononuclear Cell (PBMC) Collection

This study encompassed patients diagnosed with AP and healthy volunteers. All AP patients met the 2012 Atlanta criteria. Prior to participation, all individuals provided informed consent, which included comprehensive details regarding this study (IIT-2023-396). Briefly, peripheral blood samples (10–15 mL) were collected from both AP patients and healthy volunteers. PBMCs were isolated within two hours of collection using Ficoll density gradient (LTS10771, TBD, Beijing, China) centrifugation.

### 4.11. RT-qPCR and Western Blotting (WB)

RNA extraction was performed using the Trizol method. Reverse transcription was carried out using the Evo M-MLV cDNA Synthesis Kit (AG11307, AG, Wuhan, China), followed by RT-PCR reactions utilizing the PCR system kit (AG11707, AG, Wuhan, China), The primers are listed in Appendix A. Proteins were extracted utilizing RIPA lysis buffer (P0013, Beyotime, Haimen, China). Following centrifugation for 10 min, the concentration of the supernatant was assessed using a bicinchoninic acid (BCA) protein assay kit (A53225, Thermo Scientific, Waltham, USA). Equal amounts of the protein samples were subjected to separation via SDS (P0014D, Beyotime, Haimen, China), then transferred onto polyvinylidene difluoride (PVDF) membranes (CAS#:24937-79-9, Sigmaaldrich, St. Louis, MO, USA). These membranes were blocked with skim milk for one hour, then incubated overnight with the designated primary antibody (MGST1: ab131059, Abcam; ACSL4: ab155282, Abcam; GPX4: ab15066, Abcam; NFE2L2: ab62352, Abcam; AIFM2: ab302673, Abcam, Cambridge, UK). Afterward, the membranes were treated with secondary antibodies (PR30011, Proteintech, Rosemont, IL, USA) for one hour, and protein signals were identified using an enhanced chemiluminescence reagent (E1050, Lablead, Beijing, China), with GAPDH (60004-1-Ig, Proteintech, Rosemont, IL, USA) utilized as an internal control.

### 4.12. Immunohistochemistry (IHC)

The pancreatic tissues were fixed and subsequently embedded in paraffin. Following this, the samples were stained using a hematoxylin and eosin solution. IHC staining was performed according to standard protocols for dewaxing and rehydrating the sections. Primary antibodies (ab131059, Abcam, Cambridge, UK) were applied, followed by incubation with IgG secondary antibodies (PK10006, Proteintech, Rosemont, IL, USA). Finally, diaminobenzidine was used for staining, with hematoxylin acting as the counterstain for the sections.

### 4.13. Immunofluorescence (IF)

Paraffin-embedded tissue sections are deparaffinized with xylene and rehydrated through graded alcohols. Antigen retrieval is performed using a citrate or EDTA buffer. Sections are blocked with serum or BSA to prevent non-specific binding, then incubated with a primary antibody (ab131059, Abcam, Cambridge, UK) overnight at 4 °C. After washing, a fluorophore-conjugated secondary antibody is applied (SA00001-1, Proteintech, Rosemont, IL, USA). Nuclei can be counterstained with DAPI (BL739B, Biosharp, Wuhan, China).

### 4.14. Cell Culture and Treatment

The human normal pancreatic duct cell line H6C7 (IM-H374, Immocell, Xiamen, China) was cultured in 1640 medium with 10% fetal bovine serum, while the human pancreatic ductal adenocarcinoma cell line CFPAC-1 (IM-H156, Immocell, Xiamen, China) was grown in MEM supplemented with 10% fetal bovine serum. Both cell lines were treated with taurocholic acid sodium (TCS; 200–400 µM) (HY-N0545, MCE, Shanghai, China), a compound that mimics biliary tract injury and exhibits biological properties similar to cerulein [36,37]. For the positive control, both cell lines received 10µM RSL3 (HY-100218A, MCE, Shanghai, China), a known ferroptosis inducer. To inhibit ferroptosis, liproxstatin-1 (Lipro-1) (HY-12726, MCE, Shanghai, China) was administered at concentrations ranging from 1 µM to 10 µM 24 h prior to TCS or RSL3 treatment.

### 4.15. Cell Transfection

To determine the functions of the MGST1 gene, H6C7 cells were cultured for a 24 h period before being transfected to silence or overexpress the gene. Gene silencing was achieved using transfected 75 pmol siRNA (Genephama, Suzhou, China) with the sequence AAUACAGGAGGCCAAUUCCTT, while plasmids containing MGST1-DNA (Genephama, Suzhou, China) were used for overexpression. H6C7 cells were transfected with either a siNC or an empty vector as a negative control. For the transfection process, the Lipofectamine 3000 kit (L3000008, Invitrogen, Carlsbad, USA) was employed. Following a 48 h transfection period, the cells received additional treatments as specified in this study.

### 4.16. Enzyme-Linked Immunosorbent Assay (ELISA)

To measure MGST1 levels in cell supernatants using ELISA, microplate wells were coated with anti-MGST1 antibodies and non-specific sites were blocked (RE1837H, reedbiotech, Guildford, UK). Cell supernatants were added to the wells and incubated. Wells were washed to remove unbound components, followed by the addition of enzyme-conjugated detection antibodies. After incubation, the wells were washed again, and a substrate was added to develop the signal. Absorbance was measured at the specified wavelength to quantify MGST1.

### 4.17. CCK-8, GSH/GSSG Ratio, and ROS Assays

Cell viability was determined through the use of the CCK8 assay (C0038, Beyotime, Haimen, China). H6C7 cells were cultured in 96-well plates and treated with stimuli. Afterward, they underwent a 1 to 2 h incubation at 37 °C with the CCK8 reagent. The glutathione/glutathione disulfide (GSH/GSSG) ratio (G263, Dojindo, Kamimashiki-gun, Japan) was typically measured using a colorimetric assay, where GSH reacted with a chromogenic probe and GSSG was detected after enzymatic reduction, reflecting the cellular redox balance. ROS probes (S0033S, Beyotime, Haimen, China) were used to detect intracellular reactive oxygen species levels.

### 4.18. Statistical Analysis

All data were shown as the mean ± standard deviation (SD). GraphPad Prism software (version 9) was used for data analysis. Differences between groups were evaluated using a *t*-test or ANOVA. *p* < 0.05 was considered statistically significant.

## 5. Conclusions

In this study, we employed an integrative approach that combined various bioinformatics methodologies, multiple statistical analyses, and comprehensive investigations to elucidate the presence of ferroptosis in acute pancreatitis and identify three hub genes associated with different levels of severity. Validation was conducted using clinical samples, animal models, and in vitro experiments. We discovered for the first time that ferroptosis occurs in human pancreatic ductal cells during the disease state, with MGST1 playing a key negative feedback role in regulating ferroptosis in these cells, providing novel targets for the treatment of AP.

## Figures and Tables

**Figure 1 ijms-26-01899-f001:**
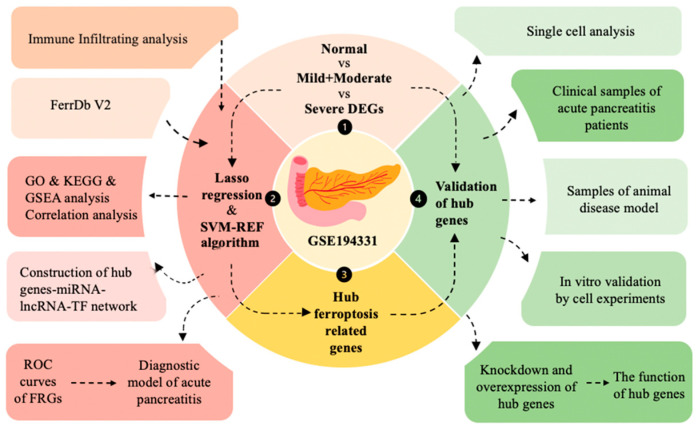
The flow chart of this study.

**Figure 2 ijms-26-01899-f002:**
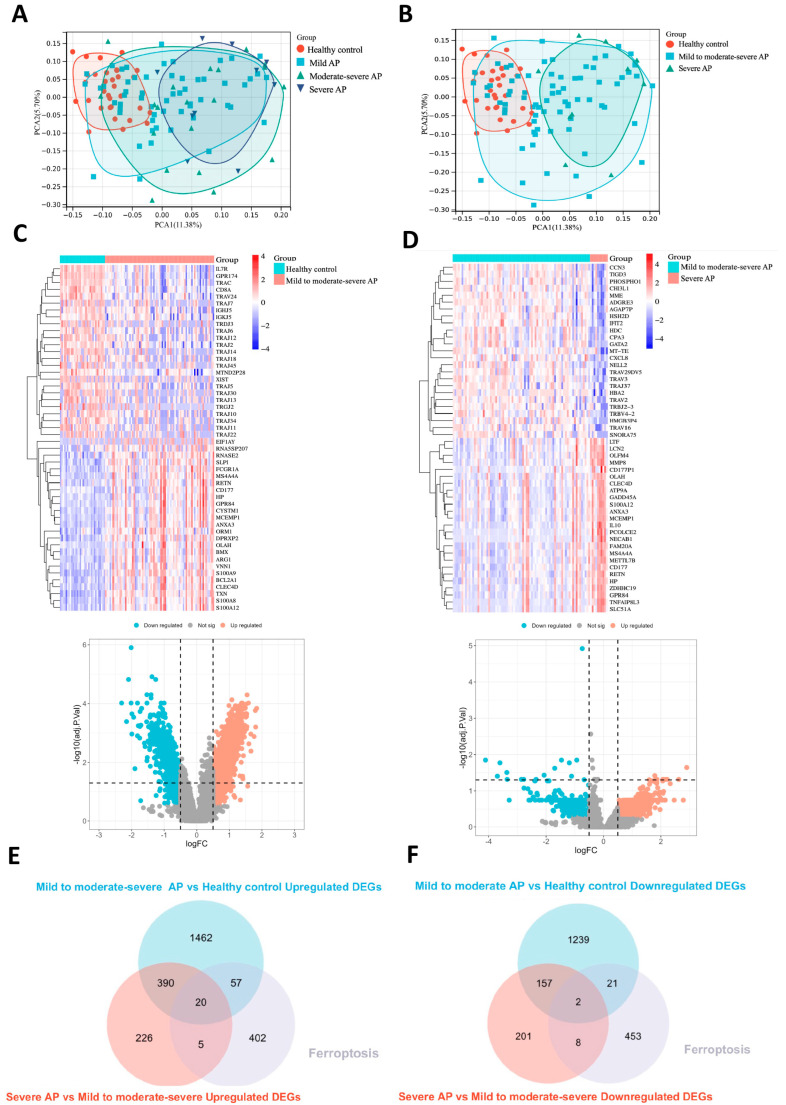
Identification of DEGs and Ferroptosis-Related Driving Genes in Acute Pancreatitis. (**A**) PCA plot showing the distribution of samples from four groups, including healthy controls, mild AP, moderately severe AP, and severe AP. The first principal component (PC1) accounts for 11.38% of the variance, while the second principal component (PC2) explains 5.7%. (**B**) PCA plot showing the distribution of samples from three groups, including healthy controls, mild to moderately severe AP, and severe AP. The first principal component (PC1) accounts for 11.38% of the variance, while the second principal component (PC2) explains 5.7%. (**C**) The heatmap of the top 50 most differentially expressed genes, including the 25 most upregulated and 25 most downregulated genes, along with the volcano plot depicting all differentially expressed genes between the healthy control and mild to moderately severe AP groups. (**D**) The heatmap of the top 50 most differentially expressed genes, including the 25 most upregulated and 25 most downregulated genes, along with the volcano plot depicting all differentially expressed genes between the mild to moderately severe AP and severe AP groups. (**E**) The Venn diagram illustrates that by intersecting 410 DEGs that were upregulated with increasing disease severity and the ferroptosis gene set, 20 ferroptosis-related genes were identified. (**F**) The Venn diagram illustrates that by intersecting 159 DEGs that were downregulated with increasing disease severity and the ferroptosis gene set, 2 ferroptosis-related genes were identified.

**Figure 3 ijms-26-01899-f003:**
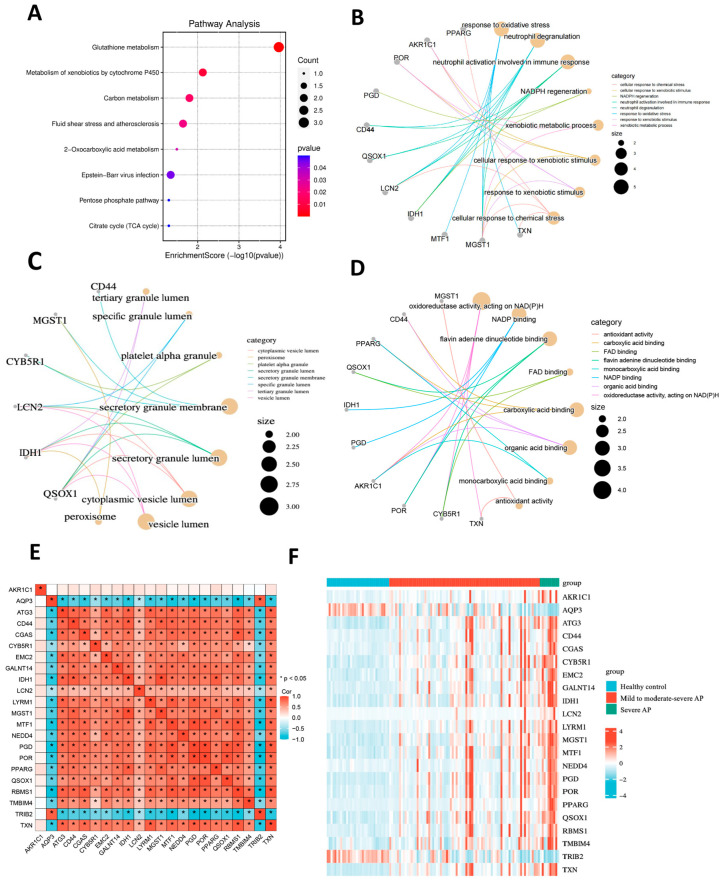
Enrichment Analysis of Ferroptosis-Related Driving Genes and Correlation Analysis. (**A**) KEGG pathway analysis of 22 ferroptosis-related driver genes. (**B**–**D**) GO enrichment analysis identified the top 10 significantly enriched pathways in BP, CC, and MF for the 22 ferroptosis-related driver genes. For all enriched GO and KEGG terms, *p* < 0.05. (**E**) The correlation heatmap of the 22 ferroptosis driver genes, * *p* < 0.05. (**F**) Heatmap of the expression levels of the 22 ferroptosis driver genes across the healthy control group, the mild to moderately severe AP group, and the severe AP group.

**Figure 4 ijms-26-01899-f004:**
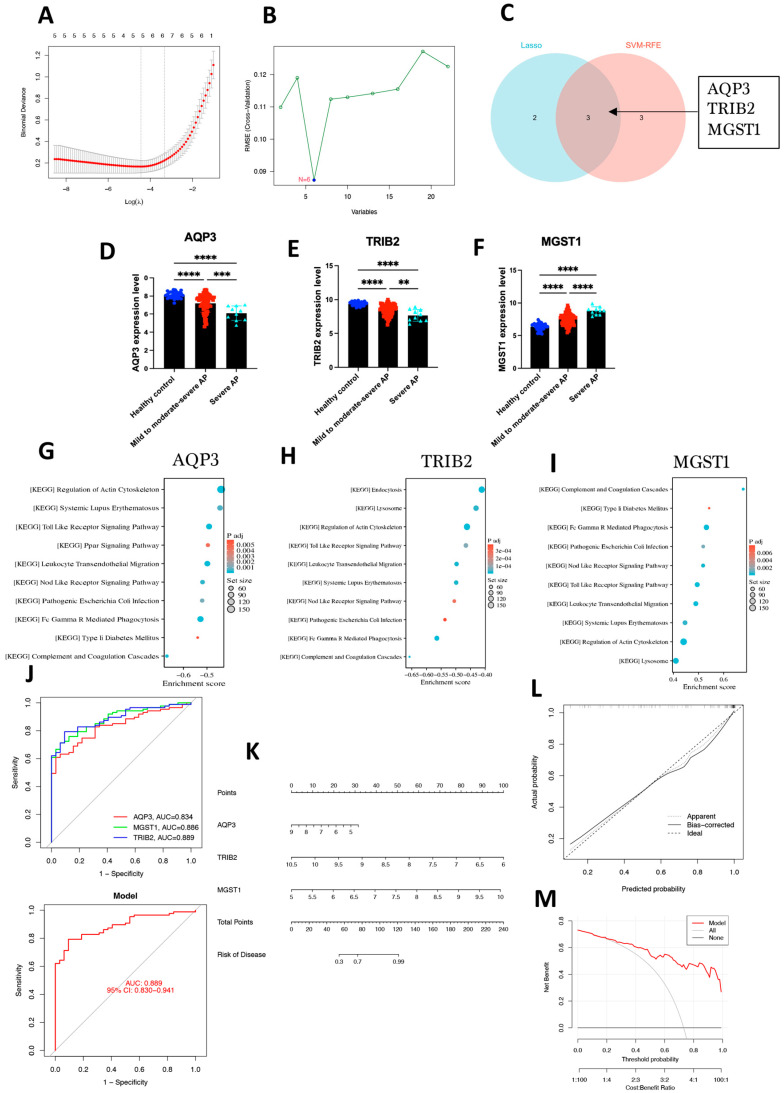
Identification of Hub Genes Using Machine Learning Methods and Construction of a Diagnostic Model. (**A**) The LASSO regression cross-validation curve illustrates the selection of optimal λ values using 10-fold cross-validation. (**B**) The SVM analysis demonstrated effective classification of samples based on ferroptosis-related gene expression profiles. (**C**) The Venn diagram shows the intersection of genes selected by Lasso and SVM algorithms, identifying three hub genes: *AQP3*, *TRIB2,* and *MGST1*. (**D**–**F**) The expression levels of *AQP3*, *TRIB2*, and *MGST1* across different groups, including healthy controls, mild to moderately severe AP, and severe AP. ** *p* < 0.01, *** *p* < 0.001, **** *p* < 0.0001. (**G**–**I**) The GSEA of the three hub genes revealed the top 10 significantly enriched pathways. (**J**) The ROC curve analysis of three hub genes. (**K**) The nomogram integrating the hub genes. (**L**,**M**) The calibration curve illustrates the connection between predicted and observed probabilities. The ideal dashed line represents perfect prediction accuracy. The visible dashed line corresponds to the whole dataset, and the bias-corrected solid line, derived through bootstrapping, reflects the actual performance of the nomogram.

**Figure 5 ijms-26-01899-f005:**
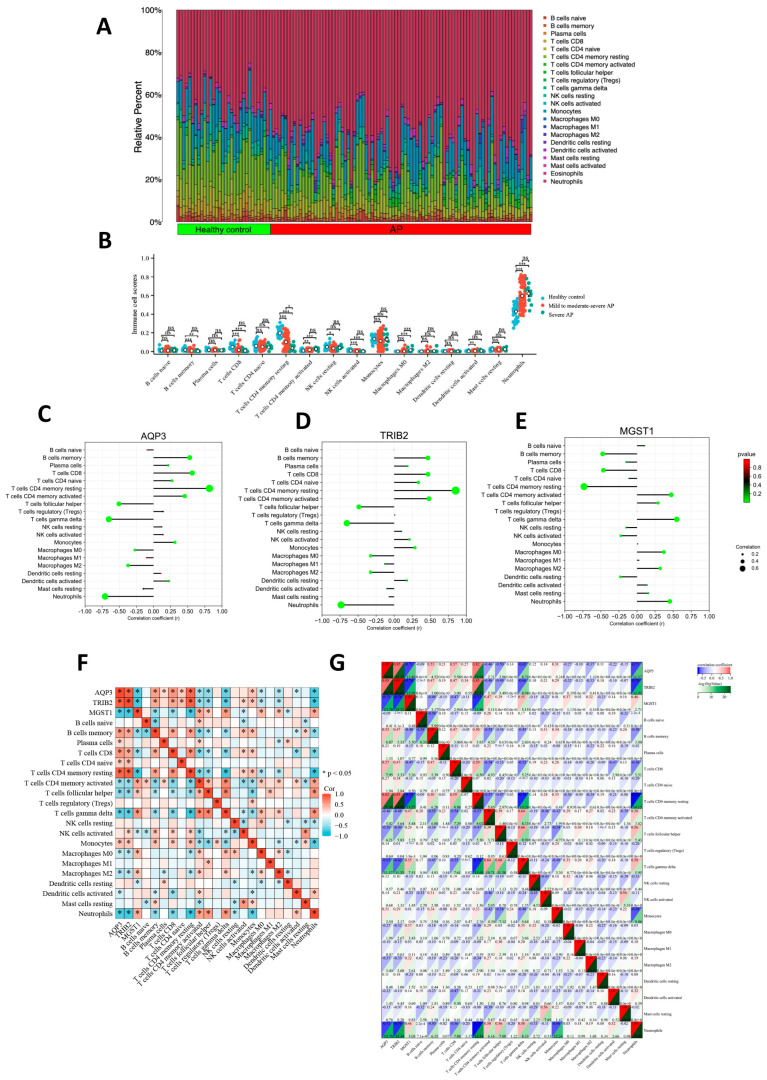
Association between Hub Genes and Immune Cell Infiltration. (**A**) The proportion distribution of 22 immune cell subtypes between the AP patients and the healthy control group. (**B**) The alterations in 16 types of immune cells across varying disease severities were assessed using the CIBERSORT algorithm. ns: not significant, * *p* < 0.05, ** *p* < 0.01, *** *p* < 0.001. (**C**–**E**) Lollipop plots were used to visualize the correlation between the three hub genes and various immune cell types. (**F**) The correlation heatmap displaying the relationships between the three hub genes and various immune cells, * *p* < 0.05. (**G**) The correlation coefficients and corresponding *p*-values between the three hub genes (*AQP3*, *TRIB2*, and *MGST1*) and various immune cells.

**Figure 6 ijms-26-01899-f006:**
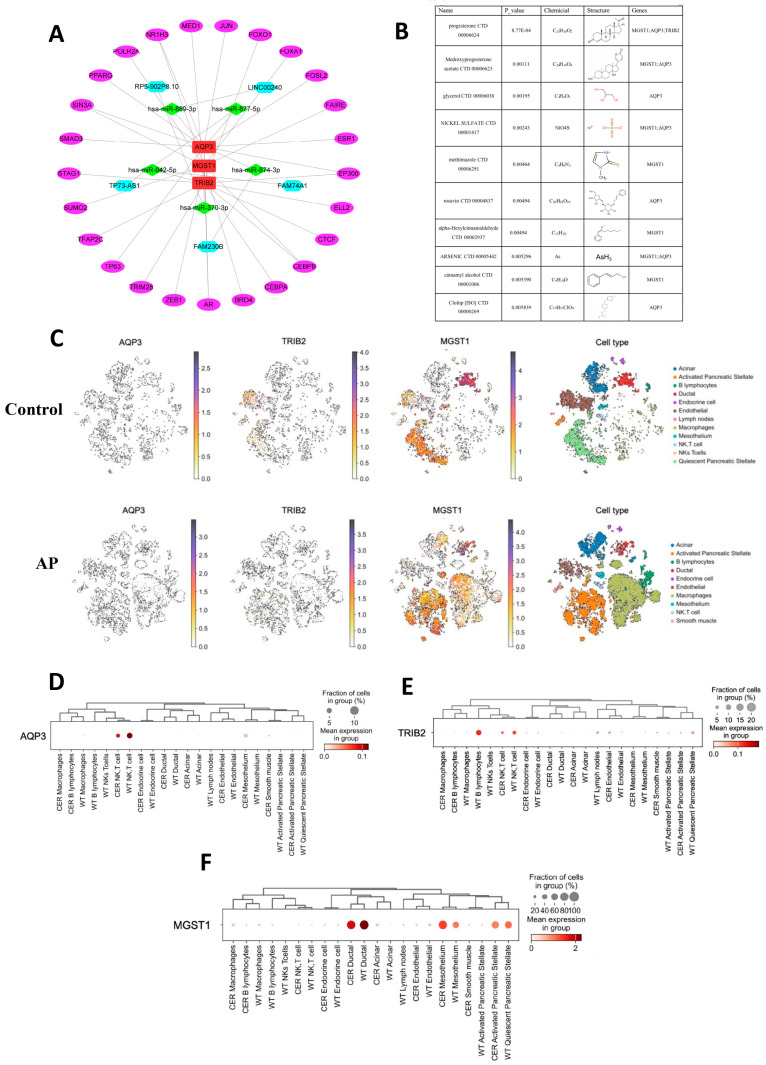
Potential Roles of the Three hub Genes in Pancreatic Tissue Inferred Through Single-Cell Analysis. (**A**) Network analysis was used to gather interactions between miRNAs, lincRNAs, TFs, and the three hub genes. The purple icons represent transcription factors, the blue icons represent lncRNAs, the green icons represent miRNAs, and the red icons represent the three hub genes. (**B**) The Drug Gene Interaction Database was used to identify drugs with strong interaction scores associated with *AQP3*, *TRIB2*, and *MGST1*. (**C**) After t-SNE processing of single-cell data from the Control and AP mouse groups, the distribution of the three hub genes across 12 different cell types within the tissue is illustrated. (**D**–**F**) The bubble plot illustrates the expression enrichment of the three hub genes across different cell types in mouse pancreatic tissue.

**Figure 7 ijms-26-01899-f007:**
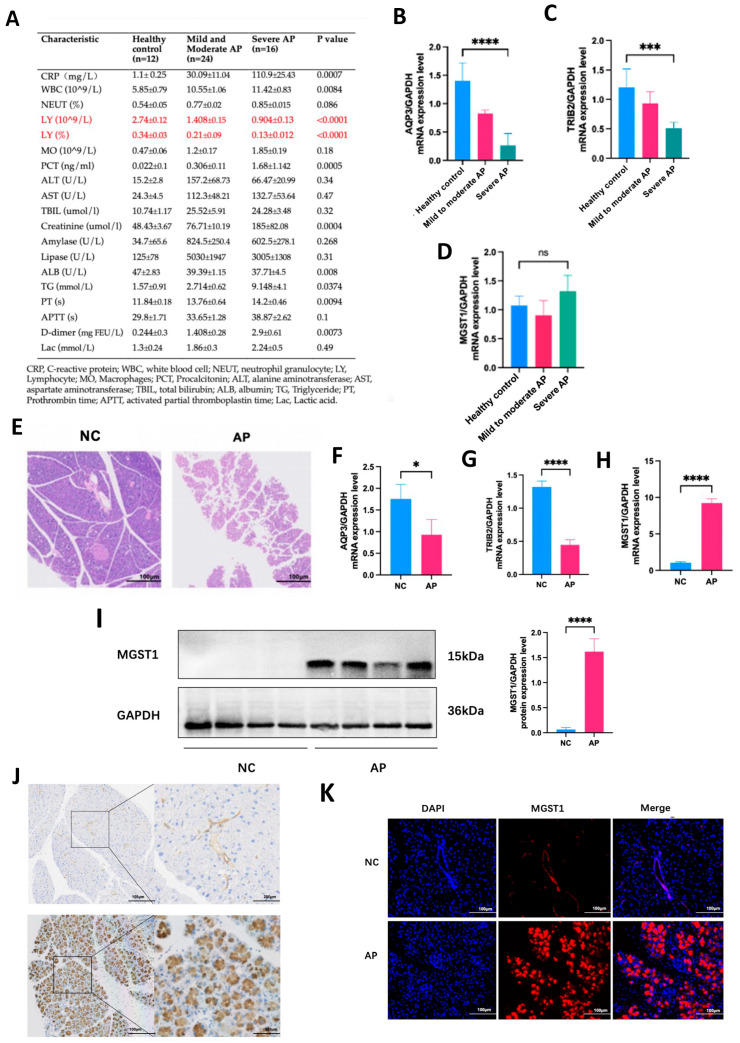
Validation of Hub Genes through Clinical Samples and Animal Models. (**A**) Clinical baseline data of 12 healthy controls, 24 mild to moderate AP patients, and 16 severe AP patients who were included in this study. (**B**–**D**) RT-qPCR results of the three hub genes in PBMCs from the patients included in this study. (**E**) HE staining of mouse pancreatic tissue within NC and AP groups. (**F**–**H**) The differential expression of hub genes in mouse pancreatic tissues between NC and AP groups was analyzed using RT-qPCR. (**I**) WB results show the changes in MGST1 expression in pancreatic tissue from NC and AP mice. (**J**,**K**) Immunofluorescence and immunohistochemistry results display the expression changes of MGST1 in pancreatic tissue from NC and AP mice. N = 4, means ± SD, * *p* < 0.05, *** *p* < 0.001, **** *p* < 0.0001.

**Figure 8 ijms-26-01899-f008:**
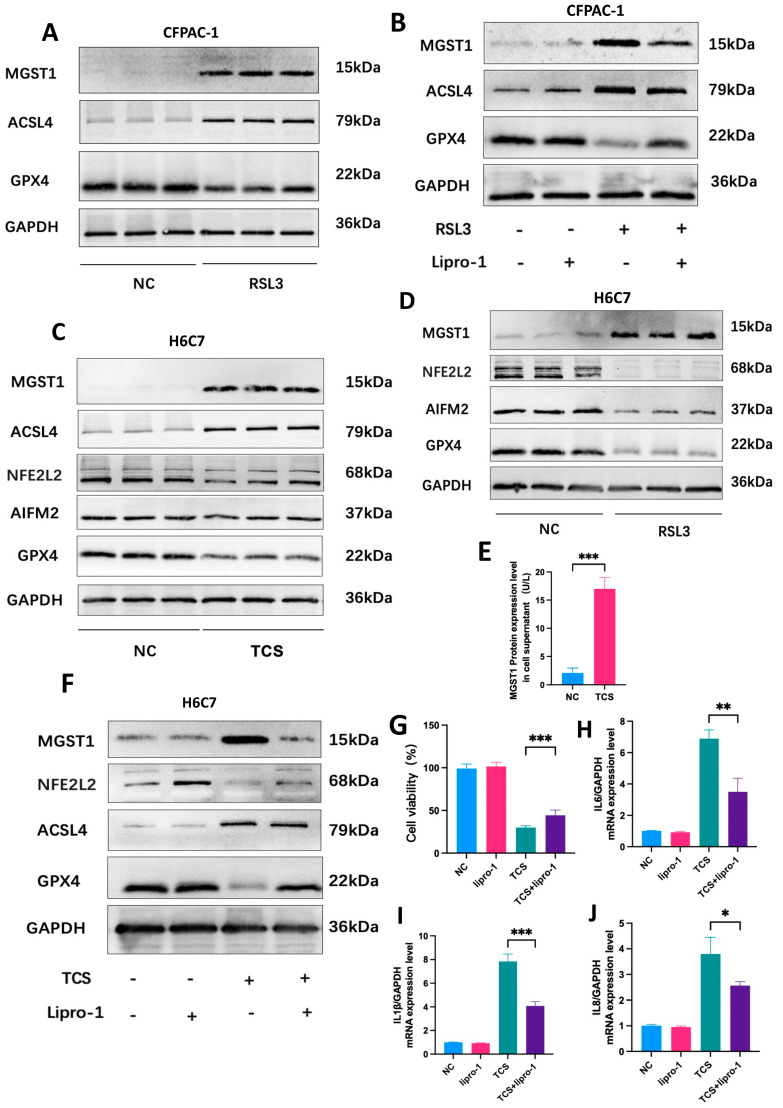
MGST1 is Upregulated during Ferroptosis in Human Pancreatic Duct Cells. (**A**) MGST1 was upregulated in response to RSL3 (10 µM; 24 h) along with activation of the ferroptosis pathway in CFPAC-1 cells. (**B**) The effects were reversed following a 24 h pretreatment with ferroptosis inhibitors (10 µM liproxstatin-1). (**C**) MGST1 protein levels were significantly elevated in H6C7 cells in response to TCS stimulation (400 µM; 24 h), along with activation of the ferroptosis pathway. (**D**) MGST1 levels also significantly increased in H6C7 cells stimulated by RSL3 (10 µM; 24 h). (**E**) ELISA results show a significant increase in MGST1 protein expression in the cell supernatant from the TCS group compared to the NC group. (**F**) The effects were reversed following 24 h pretreatment with ferroptosis inhibitors (1 µM liproxstatin-1). (**G**) After a 24 h pretreatment with a ferroptosis inhibitor (1 µM liproxstatin-1), cell viability in the TCS group significantly improved. (**H**–**J**) After 24 h pretreatment with a ferroptosis inhibitor, the expression levels of IL-6, IL-1β, IL-8, and IL-18 were significantly reduced. N = 3, means ± SD, * *p* < 0.05, ** *p* < 0.01, *** *p* < 0.001.

**Figure 9 ijms-26-01899-f009:**
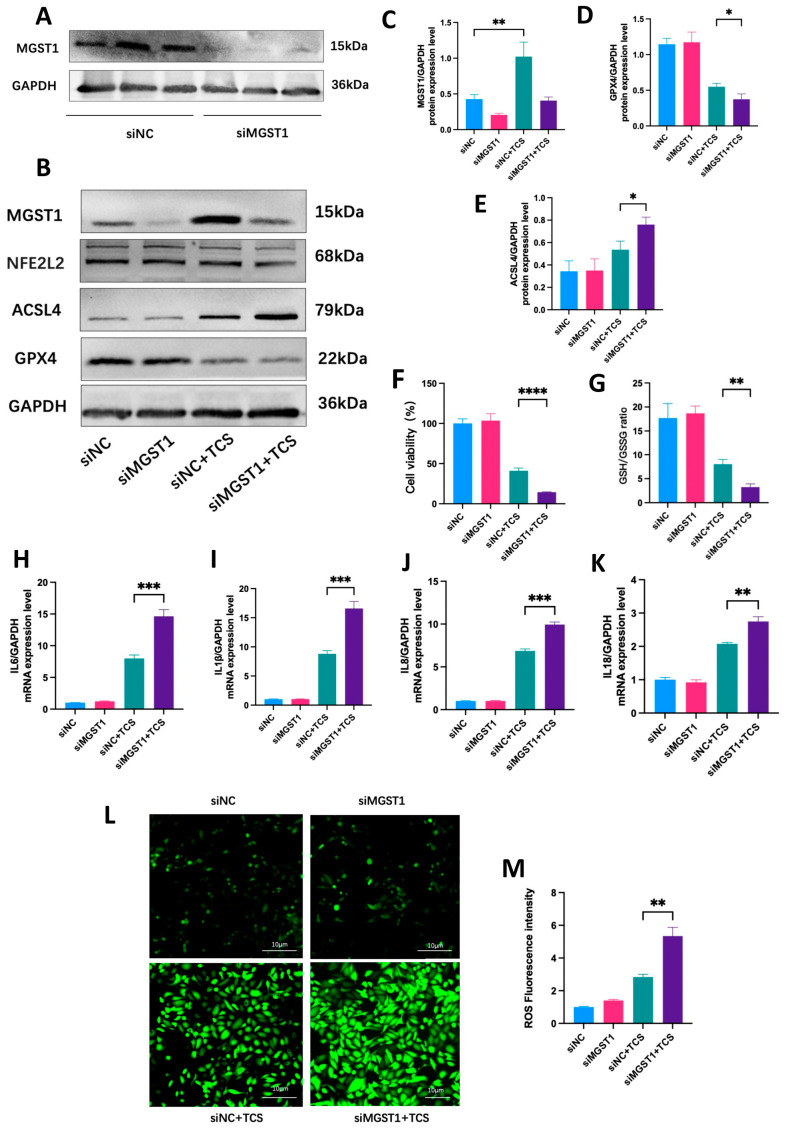
MGST1 Negatively Regulates Ferroptosis and Protects Human pancreatic Ductal Cells from Inflammatory Damage. (**A**) WB analysis showed that MGST1 protein levels in H6C7 cells were significantly decreased following transfection with siRNA. (**B**) ACSL4 expression was upregulated while GPX4 expression was downregulated in the disease group following MGST1 knockdown using siRNA; (**C**–**E**) Protein quantitative analysis of MGST1, ACSL4, and GPX4; (**F**,**G**) Cell viability and GSH/GSSG levels significantly decreased after MGST1 knockdown; (**H**–**K**) The mRNA levels of various inflammatory factors, including *IL-6*, *IL-1β*, *IL-8*, and *IL-18*, were significantly increased in the disease group following MGST1 knockdown; (**L**,**M**) ROS fluorescence intensity increased markedly after MGST1 knockdown. N = 3, means ± SD, * *p* < 0.05, ** *p* < 0.01, *** *p* < 0.001, **** *p* < 0.0001.

**Figure 10 ijms-26-01899-f010:**
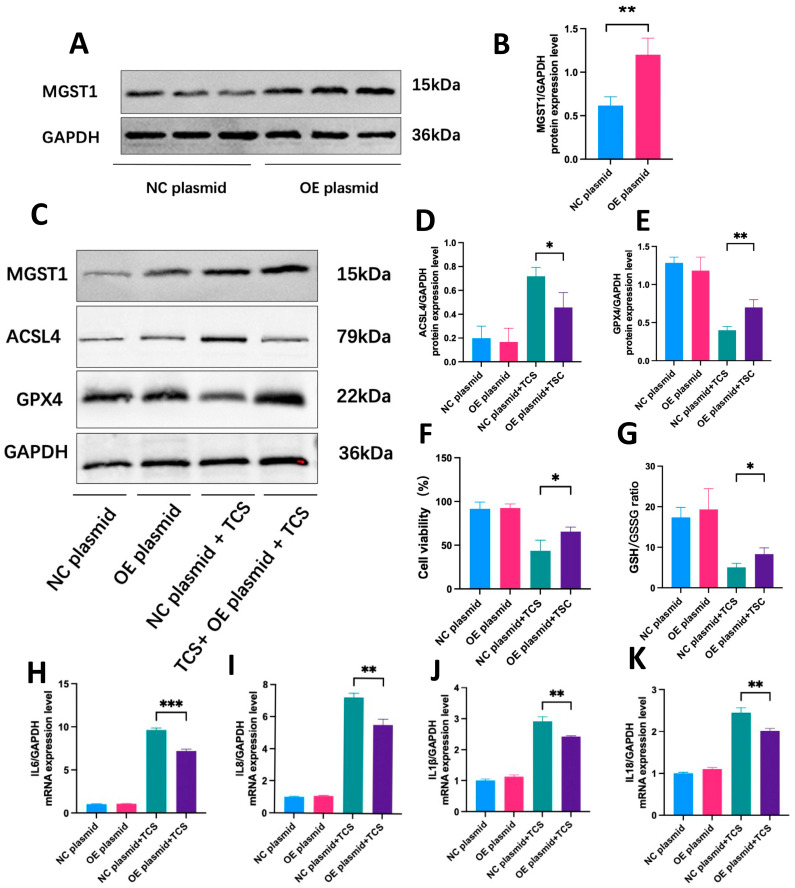
MGST1 Overexpression Alleviates Ferroptosis and Inflammatory Damage in Ductal Cells. (**A**,**B**) The protein level of MGST1 was significantly increased after transfection with the overexpressed plasmid. (**C**) MGST1 overexpression via plasmid transfection led to downregulation of ACSL4 expression and upregulation of GPX4 expression in the disease group. (**D**,**E**) Protein quantitative analysis of MGST1, ACSL4, and GPX4. (**F**,**G**) Cell viability and GSH/GSSG levels significantly increased after MGST1 overexpression. (**H**–**K**) The mRNA levels of various inflammatory factors, including *IL-6*, *IL-1β*, *IL-8*, and *IL-18*, were significantly decreased in the disease group following MGST1 overexpression. N = 3, means ± SD, * *p* < 0.05, ** *p* < 0.01, *** *p* < 0.001.

**Table 1 ijms-26-01899-t001:** The data source of this study.

Datasets	Data Source	Species	Sampling Site	Sample Grouping	Sequencing Type	Sequencing Platform	Dataset Retrieval Date
GSE194331	GEO	Homo	Peripheral blood	Healthy control = 32, Mild AP = 57, Moderately Severe AP = 20, Severe AP = 10	RNA-seq	GPL16791	2024-1-1
GSE188819	GEO	Mouse	pancreatic tissue	Control = 2, AP = 2	Single-cell-seq	GPL19057	2024-1-1
484 Ferroptosis-related genes	FerrDb V2	Homo					2024-1-1

## Data Availability

All the raw data disclosed in this article can be accessed upon request from the authors.

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
