# Peer review of "MGST1 Protects Pancreatic Ductal Cells from Inflammatory Damage in Acute Pancreatitis by Inhibiting Ferroptosis: Bioinformatics Analysis with Experimental Validation"

_ijms, 2025, doi:10.3390/ijms26051899_

Round 1

Reviewer 1 Report

Comments and Suggestions for Authors

Comments and Suggestions:

Title: MGST1 Protects Pancreatic Ductal Cells from Inflammatory Damage in Acute Pancreatitis by Inhibiting Ferroptosis: Bioinformatics Analysis with Experimental Validation.

The manuscript by Zhang et. al., describes about MGST1 which inhibits ferroptosis and protects pancreatic ductal cells from inflammatory damage in Acute Pancreatitis (AP). They used GEO datasets for identifying deregulated DEGs in increasing disease severity and found 22 ferroptosis-related driver genes. Using machine learning approach, they identified 3 hub genes AQP3, TRIB2, and MGST1 with combined AUC of 0.889. They validated the results using other datasets and in vitro experiments which discovered for first time that ferroptosis occurs in pancreatic duct cells during AP. They found that upregulation of MGST1 negatively regulates ferroptosis via ACSL4/GPX4 axis. They concluded that the results show the early diagnosis and treatment of AP.

The manuscript is well executed in the latter part, but the figure 2 has some serious flaws which need to be corrected. The preliminary analysis is incorrect and the further downstream results needs thorough modification.

Major Points:

1.      Methods: The two datasets used, one for discovery set and other for validation. Is it feasible that hub genes from human can be validated in mouse datasets? GEO database contains several other datasets related to acute pancreatitis from human source. These datasets can also be used for validation. Why the authors chose the mouse dataset?

2.      Methods:  Also, only two samples from each group of validation set is sufficient enough to validate the findings?

3.      Line 252-252: the percentage of principal components are not matching with the figure 2A.

4.      Figure 2B: why the figure is showing here 4 groups and in figure 2A three groups?

5.      Figure 2C, 2D: Similarly, the heatmaps also showed only two groups. They can be combined in a single heatmap to make it more informative.

6.      Volcano plot: the cutoff for logFC in methods is >1. But the cutoff shown in the figure as lines does not match. How the authors identified 641 upregulated and 366 downregulated genes between the mild to moderately-severe AP and the severe AP groups, because in volcano plot 2, there are only few genes above the cutoff values, then how they identified these many genes. The results in the figure 2 are too much confusing. Please revise thoroughly.

7.      Table 1, line 88: the authors should mention the date of data taken from database. Also, there are 431 taken for analysis, but in figures 2E and 2F, the total number of ferroptosis related genes are 484 in both venn diagrams.

Minor Points:

1.      Figure 2A: the PCA plot should show 4 different groups based on the information provided in the dataset to confirm the significant separation among all groups.

2.      Figure 2C, line 229-231: the words can be rearranged as mild to moderately-severe AP vs healthy control groups. Similarly, severe AP vs mild to moderately-severe AP groups. Also, the group name should be same throughout the manuscript. e.g. normal or healthy control groups.

Author Response

Please see the attachment. We appreciate your thorough review.

Reviewer 2 Report

Comments and Suggestions for Authors

The manuscript by Zhang et al. investigates the key ferroptosis regulators in AP using bioinformatics, single-cell analysis, animal models, and in vitro experiments. They identify AQP3, TRIB2, and MGST1 as key regulators, particularly MGST1, which protects pancreatic ductal cells from inflammatory damage in acute pancreatitis (AP) by inhibiting ferroptosis via the ACSL4/GPX4 axis. This research advances our understanding of ferroptosis in AP and provides a promising foundation for future therapeutic investigations. Overall, while the manuscript is logical, its writing and figures could be improved.

I have only one comment: the authors should provide the full name of each gene upon its first appearance.

Comments on the Quality of English Language

the manuscript is logical, its writing and figures could be improved.

Author Response

Please see the attachment.We appreciate your thorough review.

Round 2

Reviewer 1 Report

Comments and Suggestions for Authors

The comments have been addressed and resolved significantly. The manuscript is now updated with the required data and figures and it is deemed suitable for publication in IJMS.